# Noninvasive Measurement of Time-Varying Arterial Wall Elastance Using a Single-Frequency Vibration Approach

**DOI:** 10.3390/s20226463

**Published:** 2020-11-12

**Authors:** Jia-Jung Wang, Shing-Hong Liu, Wei-Kung Tseng, Wenxi Chen

**Affiliations:** 1Department of Biomedical Engineering, I-Shou University, Kaohsiung 824, Taiwan; wangjj@isu.edu.tw; 2Department of Computer Science and Information Engineering, Chaoyang University of Technology, Taichung 413, Taiwan; 3Department of Cardiology, E-Da Hospital, Kaohsiung 824, Taiwan; tsengarthur@gmail.com; 4Biomedical Information Engineering Laboratory, The University of Aizu, Aizu-Wakamatsu City, Fukushima 965-8580, Japan; wenxi@u-aizu.ac.jp

**Keywords:** vibrator, force sensor, wall elastance, arterial stiffness, single-frequency vibration approach

## Abstract

The arterial wall elastance is an important indicator of arterial stiffness and a kind of manifestation associated with vessel-related disease. The time-varying arterial wall elastances can be measured using a multiple-frequency vibration approach according to the Voigt and Maxwell model. However, such a method needs extensive calculation time and its operating steps are very complex. Thus, the aim of this study is to propose a simple and easy method for assessing the time-varying arterial wall elastances with the single-frequency vibration approach. This method was developed according to the simplified Voigt and Maxwell model. Thus, the arterial wall elastance measured using this method was compared with the elastance measured using the multiple-frequency vibration approach. In the single-frequency vibration approach, a moving probe of a vibrator was induced with a radial displacement of 0.15 mm and a 40 Hz frequency. The tip of the probe directly contacted the wall of a superficial radial artery, resulting in the arterial wall moving 0.15 mm radially. A force sensor attached to the probe was used to detect the reactive force exerted by the radial arterial wall. According to Voigt and Maxwell model, the wall elastance (E_single_) was calculated from the ratio of the measured reactive force to the peak deflection of the displacement. The wall elastances (E_multiple_) measured by the multiple-frequency vibration approach were used as the reference to validate the performance of the single-frequency approach. Twenty-eight healthy subjects were recruited in the study. Individual wall elastances of the radial artery were determined with the multiple-frequency and the single-frequency approaches at room temperature (25 °C), after 5 min of cold stress (4 °C), and after 5 min of hot stress (42 °C). We found that the time-varying E_single_ curves were very close to the time-varying E_multiple_ curves. Meanwhile, there was a regression line (E_single_ = 0.019 + 0.91 E_multiple_, standard error of the estimate (SEE) = 0.0295, *p* < 0.0001) with a high correlation coefficient (0.995) between E_single_ and E_multiple_. Furthermore, from the Bland–Altman plot, good precision and agreement between the two approaches were demonstrated. In summary, the proposed approach with a single-frequency vibrator and a force sensor showed its feasibility for measuring time-varying wall elastances.

## 1. Introduction

The relationship between arterial stiffness and cardiovascular disease has been extensively studied in the last 10 years. Arterial stiffness increases with both age and disease states associated with cardiovascular risks, including hypertension, diabetes mellitus, hypercholesterolemia, and end-stage renal failure [1,2]. When the change in arterial stiffness is clinically detected before the appearance of vascular diseases, it can either be as a marker for the development of future atherosclerotic disease or more directly involved in the process of atherosclerosis [3,4]. Moreover, the arterial stiffness is an important parameter when remodeling coronary and peripheral arteries [5,6]. Consequently, early detection of a change in arterial stiffness before the clinical emergence of vascular illness may serve as a valuable prognostic indication for cardiovascular disease and other chronic diseases.

Many physical parameters of arterial stiffness have been proposed to assess the global arterial stiffness, such as the pulse pressure [7], pulse wave velocity [8], the capacity compliance of the aorta [9], augmentation index [10], carotid intima-media thickness [11], and β variable [12]. In addition, various indices have been submitted to appraise the local arterial stiffness, such as the compliance of brachial artery [13], arterial distensibility [14], spring constant of the arterial wall [15], Young’s elastic modulus [16], and arterial wall elastance [17]. These techniques for measuring arterial stiffness commonly have their own advantages and disadvantages. In the echocardiographic technique [18], although an absolute value of vascular diameter can be accurately detected, the physician’s professional training, clinical experience, and technical skills in the operation of ultrasonography equipment may affect its accuracy and reliability. Photoplethysmography can continuously detect change in blood volume because red blood cells have a similar light absorption coefficient [19]. The number of red cells depends on the blood volume. However, the greatest disadvantage of this method is the difficulty in accurately calibrating the changed volume. Meanwhile, a similar drawback exists in impedance plethysmography, in which the change in vascular volume can be detected due to different electrical characteristics of tissue [20]. The tonometry of a liquid or an air chamber is used to indirectly sense the change in arterial volume [21]. A cuff model built using a flow sensor and pressure sensor is used to transfer the amplitude of pulse pressure to the amplitude of pulse volume [22]. However, these methodologies have a common disadvantage regarding how to effectively and correctly transfer the detected physical quantities to the actual change in arterial volume. Some studies applied more complicated schemes to approximately calculate the arterial stiffness. In the analysis of pulse transit time [23], the arterial stiffness was estimated by assuming the wall thickness and lumen radius of arteries to be constant in the Moen–Korteweg relationship. In the analysis of pressure waveform [24], the exponentially decayed coefficient during diastolic duration was found to be associated with the arterial compliance, and it could be utilized to measure the arterial stiffness.

Most of the techniques mentioned above can roughly provide an index of arterial stiffness for either an arterial segment or a whole arterial system. However, the arterial characteristics, especially in terms of arterial compliance and wall elastance, are not constant in essence but time-varying and dependent on the transmural pressure [8,17,22]. The Voigt and Maxwell model has been used to measure the time-varying arterial wall elastance with a multiple-frequency vibrator approach [17]. Because our method measures the time-varying arterial wall elastance with a simplified Voigt and Maxwell model [25], the measured elastances were compared with those obtained using the multiple-frequency vibration approach. Different vibrational frequencies were employed to evaluate the performance of this proposed approach. Moreover, to intentionally alter the arterial wall elastance, we measured the arterial wall elastance under cold- and hot-stress trials. Twenty-eight healthy and young subjects were recruited to participate in these experiments. It was found that a lower vibrational frequency resulted in a closer arterial wall elastance measurement using the multiple-frequency vibrator approach. Therefore, the purposes of our study were: (1) to present the derivation of a new method for measuring arterial wall elastance with a single-frequency vibration, (2) to compare the elastance values measured with the single- and the multiple-frequency vibration approaches in three thermal conditions, and (3) to validate the feasibility of the proposed approach by comparing its precision and agreement with the multiple-frequency approach.

## 2. Materials and Methods

### 2.1. Vascular Wall Model

A three-component lumped model of an arterial segment, modified according to the Voigt and Maxwell model [25], was established to characterize the mechanical and viscoelastic properties of the arterial wall in the study. In the lumped model, the three components were arranged in parallel and represented the elastance (E), the acting mass (M) (or inertia), and the viscosity (η) of the arterial wall, as shown in Figure 1. When the outside of the arterial wall was stimulated using a vibrator to produce a sinusoidal displacement, X(t), radially or perpendicular to the axis of an arterial lumen, the arterial wall would give off a reactive force, F(t), in response to the radial displacement. In accordance with the force balance theory on the arterial wall, the reactive force should be equal to the summation of the elastance-related force, inertia-related force, and viscosity-related force [26]. As expressed in Equation (1), the inertia-related force, the first term on the right side, is the product of the effective mass M and the acceleration of the arterial wall. Similarly, the viscosity-related force, the second on in the right side, is the product of the wall viscosity η and the velocity of the arterial wall. Lastly, the elastance-related force, the third term on the right side, is the product of the wall elastance E(t) and the displacement of the arterial wall. In this model, the elastance component was reasonably assumed to be a function of time and transmural pressure.
(1)$$F\left(t\right)=M\frac{d^{2}X\left(t\right)}{{\mathrm{dt}}^{2}}+\eta \frac{\mathrm{dX}\left(t\right)}{\mathrm{dt}}+E\left(t\right)X\left(t\right)$$

The radial displacement, X(t), is described by a sinusoidal wave, while D is the maximum shift and ω is an angle frequency. Thus, it can be expressed as
(2)X(t)=Dsin(ωt)

Then, the first and second derivatives of X(t) can be expressed using Equations (3) and (4), respectively.
(3)$$\frac{\mathrm{dX}\left(t\right)}{\mathrm{dt}}=D\omega \cos(\omega t)$$
(4)$$\frac{d^{2}X\left(t\right)}{\mathrm{dt}}=-{D\omega }^{2}\sin(\omega t)$$

After appropriate substitution of Equations (2) to (4), Equation (1) can be rearranged as follows:(5)$$\frac{F\left(t\right)}{D}=\left[E\left(t\right)-{M\omega }^{2}\right]\sin(\omega t)+\eta \omega \cos(\omega t)$$

When sin(ωt) = 1 and cos(ωt) = 0 at a specific time, T_m_, Equation (5) can be simplified as follows:(6)$$\frac{F_{T}}{D}=-{M\omega }^{2}+E_{T}$$
where E_T_ = E(T_m_) and F_T_ = F(T_m_).

### 2.2. Multiple-Frequency Estimation of Wall Elastance

Equation (6) is basically a linear function with a slope, −M, and a constant, $E_{T}$. Figure 2 shows the relation between F_T_/D and ω^2^ at a specific time, T_m_, when F_T_/D and ω^2^ are considered as the dependent and independent variables. The procedures for measuring the wall elastance (E_multiple_) with the multiple-frequency vibration were introduced in our previous study [17]. In this section, we briefly describe the related process. The frequency of the vibrator was increased from 40 to 85 Hz at a step of 5 Hz (10 frequencies in total). For each of the 10 frequencies, we could yield the F_T_/D value at a specific time which was defined within one cardiac cycle. There were 12 timings in one cardiac cycle. Thus, we could obtain 10 F_T_/D values with respect to those 10 frequencies at a specific time. Then, a regression line (red) could be constructed on the basis of the 10 points. Figure 2 shows the regression, where E_T_ is equal to the intercept of the vertical axis, M is the absolute slope of the regression line, and T_m_ is one of the 12 specific timings.

As is known, the wall elastance varies with time. To construct a wall elastance curve of one cardiac cycle from the reactive force signal, 12 timings (indicated by T_1_–T_12_) of the reactive force signal within one cardiac cycle were defined for each sinusoidal frequency. The cardiac cycles were simultaneously annotated using the electrocardiogram. Figure 3a shows the reactive force signal under a vibrator frequency of 40 Hz. Figure 3b shows the reactive force signal within one cardiac cycle. In the 12 specific timing points, T_1_ and T_12_ were annotated at the beginning and ending cycles of the reactive force signal, and the fifth timing section (T_5_) was annotated at the maximum-amplitude cycle of the reaction force signal within the systolic duration. The duration between T_1_ and T_5_ was evenly divided into T_2_, T_3_, and T_4_. In the same way, the duration between T_5_ and T_12_ was equally separated to yield T_6_, T_7_, T_8_, T_9_, T_10_, and T_11_. According to Equation (6), F_T_/D values at the specific timing points could be calculated as the peak force divided by the peak displacement under 10 different angular frequencies, as shown in Figure 2. Therefore, the 12 E_T_ values could be used to construct the wall elastance curve of one cardiac cycle.

### 2.3. Single-Frequency Estimation of Wall Elastance

After rearranging Equation (6), we can obtain Equation (7). At a specific time, T_m_, E_T_ in Equation (7) is determined as a summation of two terms. The first term, F_T_/D, on the right side is governed by the ratio of the maximum force, F_T_, to the maximum displacement, D. It is considered as the force-dependent elastance since it is associated with the reactive force. The second term, Mω^2^, is determined from the multiplication of the effective mass, M, by the squared angular frequency, ω^2^. It is considered as the frequency-dependent elastance since it is related to the angular frequency of the vibrator. Obviously, with a constant effective wall mass, the frequency-dependent elastance is positively proportional to the angular frequency squared. When the arterial wall is stimulated by a vibrator with a very low frequency, the frequency-dependent elastance becomes relatively small and is negligible. Thus, E_T_ can be calculated as a function of F_T_/D, as shown in Equation (8).
(7)$$E_{T}=\frac{F_{T}}{D}+{M\omega }^{2}$$
(8)$$E_{T}\approx \frac{F_{T}}{D}$$

In this study, the vibrator used a relatively low frequency (40 Hz) to stimulate the arterial wall for assessing the time-varying wall elastances.

## 3. Measurement System

The schematic diagram of the sensor arrangement in the designed measurement system is shown in Figure 4. There were two sensors, including a vibrator and a force sensor, used for measuring the radial arterial elastance. The vibrator was fixed by the author-designed frame and could drive the moving probe to carry out a sinusoidal displacement. The moving probe contained a force sensor to measure the reactive force of arterial wall. One end of the probe was used to press the radial arterial wall of interest, usually at the superficial segment of the radial artery near the wrist. A universal function generator was used to deliver a sinusoidal signal for driving the vibrator (DPS-270, DiaMidical System Cor., Tokyo, Japan). Then, the reactive force and the displacement signals of the vibrator were both acquired using the MP100 system (BIOPAC System, Inc., Goleta, CA, USA) with a sampling rate of 1000 Hz and a digital resolution of 12 bits. In the study, the maximum displacement of the vibrator was 0.15 mm and the diameter of the probe tip was 2 mm. Moreover, AcqKnowledge Software 3.9 (BIOPAC System, Inc., Goleta, CA, USA) was adopted to process the signals.

## 4. Experimental Protocol

Twenty-eight healthy subjects participated in the experiment to perform the wall elastance measurement at room temperature. The data of this measurement were used as the baseline to compare the data measured under cold stress and hot stress. Sixteen subjects were male, and 12 subjects were female. The average age of the subjects was 23 ± 3 years, their average blood pressure was 116 ± 17 mmHg for systolic pressure and 71 ± 12 mmHg for diastolic pressure, and the average heart rate was 74 ± 10 beats/min. Due to time limitation, only 23 and 21 participants performed the wall elastance measurements in cold stress and hot stress, respectively. The clinical trial was permitted by the Institutional Review Board of E-DA Hospital, Kaohsiung, Taiwan (no. EMRP61101N), and informed consent was received from each participant before the experiment. The room temperature in the research lab was kept at 25 °C. Each participant was required to sit on a chair and take at least 5 min of rest before the trial began. During the measurement, the subject’s left hand was put on the table with the wrist resting on a soft pillow and the palm pointing upward. The superficial part of the radial artery was forced by the moving probe connected to the vibrator. The frequency of the vibration was increased from 40 to 85 Hz at a step of 5 Hz. Subjects were measured at room temperature (25 °C) as the baseline data. During the cold-stress trial, a plastic bag containing a mixture of water and ice (about 4 °C) was placed on the inside surface of the left forearm. Then, the subjects took a 5 min rest to recover their hemodynamic variables. During the hot-stress trial, another plastic bag filled with hot water (about 42 °C) was put on the inside surface of the left forearm.

## 5. Statistical Analysis

The quantitative data are expressed as the mean ± SD. A two-tailed paired *t*-test was used to compare the average of the maximum wall elastances in cold stress or hot stress with that at room temperature. A *p*-value of 0.05 or less was considered statistically significant. The maximum values of time-varying wall elastances, measured using the multiple- and the single-frequency approaches are represented by E_multiple_ and E_single_, respectively. The degree of linear relationship between E_single_ and E_multiple_ was expressed with a Pearson correlation coefficient using Sigma Plot 12.0 (Systat Software, Inc., San Jose, CA, USA). The percentage difference between all paired values measured using these two approaches was calculated to further evaluate the feasibility of the proposed method. Here, the percentage difference (%) was defined as the absolute difference between E_multiple_ and E_single_ divided by E_multiple_. Furthermore, the precision of and the agreement between E_single_ and E_multiple_ were compared using a Bland–Altman plot [27].

## 6. Results

### 6.1. Comaprison of Maximum Wall Elastances Using Multiple- and Single-Frequency Approaches

At 4 °C (cold stress), 25 °C (room temperature), and 42 °C (hot stress), the averages of E_multiple_ were all larger than the averages of E_single_, as shown in Table 1. Moreover, E_single_ under cold stress was significantly greater than that at room temperature (*p* < 0.001). Furthermore, E_single_ under hot stress was significantly smaller than that at room temperature (*p* < 0.05). Likewise, a similar phenomenon occurred for the E_multiple_ measurement.

Table 2 shows the correlation between E_single_ and E_multiple_. For all three temperatures (4 °C, 25 °C, and 42 °C), very high correlation coefficients (larger than 0.99) were found between E_single_ and E_multiple_. Furthermore, we found that the slopes of the three regression lines corresponding to the different temperatures were all smaller than 1.0. Combining all values of E_single_ and E_multiple_ measured at the three different temperatures, we established a regression line (E_single_ = 0.019 + 0.913E_multiple_) with a Pearson correlation coefficient of 0.995, as shown in Figure 5.

### 6.2. Comparison of Time-Varying Wall Elastances Using Multiple- and Single-Frequency Approaches

Figure 6 demonstrates the time-varying wall elastance curves within a cardiac cycle using different vibrational frequency approaches at room temperature (25 °C) for one of the subjects. The time-varying wall elastance curve measured using the multiple-frequency method is denoted by black solid circles and a solid line within a cardiac cycle of 0.772 s. The other time-varying wall elastance curves using the single-frequency method are also shown in Figure 6. The vibrational frequencies of the single-frequency approach included 40 Hz, 45 Hz, 60 Hz, and 80 Hz. In the 40 Hz approach, the time-varying wall elastance curve is denoted by red solid circles and a dotted line with 32 data points. In the 45 Hz approach, the time-varying wall elastance curve is denoted by green inverted triangles and a long-dashed line with 36 data points. In the 60 Hz approach, the time-varying wall elastance curve is denoted by yellow triangles and a short-dashed line with 47 data points. In the 80 Hz approach, the time-varying wall elastance curve is denoted by blue rectangles and a solid line with 61 data points. It was found that the time-varying wall elastance curve measured using the 40 Hz approach closely followed that using the multiple-frequency method. However, the time-varying wall elastance curves measured using the 45 Hz, 60 Hz, and 80 Hz approaches deviated from that using the multiple-frequency vibration.

Figure 7 demonstrates the time-varying wall elastance curves within a cardiac cycle using different vibrational frequency approaches under cold stress (4 °C) for one of the subjects. The time-varying wall elastance curves measured using the multiple-frequency approach and the four single-frequency approaches are denoted as in Figure 6. It is remarkable that the distribution of time-varying wall elastance curves under cold stress was similar to the results at room temperature. Figure 8 demonstrates the time-varying wall elastance curves within a cardiac cycle using different vibrational frequency approaches under hot stress (42 °C) for one of the subjects. The time-varying wall elastance curves, measured using the multiple-frequency approach and four single-frequency approaches, are denoted as in Figure 6. It is remarkable that the distribution of time-varying wall elastance curves under hot stress was similar to the results at room temperature.

### 6.3. Bland–Altman Plot

Figure 9 shows a Bland–Altman plot exhibiting the extent of agreement among all tests, namely, those at room temperature, under cold stress, and under hot stress. In total, 72 paired points of E_single_ and E_multiple_ (28 obtained at room temperature, 23 obtained under cold stress, and 21 obtained under hot stress) are plotted in Figure 9. The mean and standard deviation (mean ± SD) were −29 ± 39 × 10^3^ dyne/cm. We found that the averages of all values were close, and all data points were within the limits of agreement, although three data points under cold stress fell outside of the limits of agreement. Most paired points within 1.0–1.4 of (E_single_ + E_multiple_)/2 were from the cold-stress test. Under these conditions (4 °C), the radial arterial segment may be constricted due to thermoregulation, resulting in a smaller diameter. Subsequently, it was difficult to precisely locate the probe on the radial artery and obtain accurate elastances measured using either the single- or the multiple-frequency approach.

## 7. Discussion

The single-frequency vibration approach proposed in this study can assess the time-varying arterial wall elastances, which can be intuitively considered as an indication of arterial stiffness. However, most previous studies used indirect or direct methods to explore the local physical parameters of arterial stiffness, such as the distensibility [14], compliance [9], spring constant [15], and elastance [17], and the global physical parameters, such as the pulse wave velocity [7] and augmentation. These parameters could be measured using photoplethysmography [28], echocardiography [18], pulse wave velocity [7], oscillometry [22], pulse wave analysis [24], and impedance plethysmography [29]. These physical parameters of arterial stiffness are all considered time-invariant variables. According to the theorem of the spring constant, the arterial wall elastance was estimated by Wei using the radial blood pressure waveform and the photoplethysmographic signal [26]. Wang et al. improved the method of Wei’s study using a multiple-frequency vibration approach to measure the time-varying wall elastance [17]. In their study, the measured arterial wall elastance featured a precise unit representing an absolute value which varied with the transmural pressure. Thus, this method could help researches to reveal the dynamic characteristics of the arterial wall.

To address the disadvantages of the multiple-frequency vibration approach, the wall elastance corresponding to a specific timing can be measured after a series of processing steps. First, 10 values of F_T_/D with respect to the same specific timing in different cardiac cycles were calculated by means of 10 different frequencies with the same peak deflection of the sinusoidal displacement. On the basis of the 10 paired points (F_T_/D versus ω^2^), a regression line was constructed through a curve-fitting procedure. The intercept of the vertical axis was defined as the arterial wall elastance for that specific timing point [17]. Compared with the multiple-frequency approach, the proposed single-frequency approach merely employs a 40 Hz vibration to assess the time-varying arterial wall elastances.

Skin circulation is controlled by thermoregulatory and non-thermoregulatory reactions [30,31]. Thermoregulatory reactions include skin blood flow and local arterial compliance responses to heat and cold stresses. Two branches of the sympathetic nervous system exert these effects, a noradrenergic vasoconstrictor branch and an active vasodilator branch. A recent study showed that cold stress causes acute decreases in central and peripheral compliance [32], whereas heat stress increases vascular compliance [33]. Therefore, in this study, cold stress and hot stress were chosen to induce an alteration in the radial arterial wall elastances. The regression line between E_single_ and E_multiple_ obtained under three different thermal conditions (4 °C, 25 °C, and 42 °C) was found to have a high correlation coefficient of 0.995, as shown in Figure 5. In addition, the distribution of 72 scattered points in the Bland–Altman plot in Figure 9 shows that most points were within the limits of agreement, suggesting good agreement between the single-frequency and multiple-frequency approaches. Moreover, in Table 1, E_single_ under both cold stress and hot stress showed significant differences with respect to that at room temperature. Moreover, among the three thermal conditions, the average of E_single_ under cold stress was largest, and the average of E_single_ under hot stress was smallest. These findings are consistent with physiological thermal stimulations.

According to Equation (7), because E_T_ is related to F_T_/D and ω^2^, its measurement must use the multiple-frequency approach. However, when we only used the 40 Hz approach to evaluate E_T_ according to Equation (8), the results of all experiments in Table 2 show that E_single_ and E_multiple_ were very close. Thus, we can consider that the effect of the force-dependent component on the wall elastance measurement is larger than that of the frequency-dependent component.

The medium between the tip of the moving probe and the radial arterial wall mainly includes the skin and muscle. Such a medium may have influenced the accuracy of the arterial wall elastance measurements in this study. As a result, the tip of the moving probe had to be correctly placed at the superficial segment of a radial artery, due to the presence of less tissue. The surface area on the superficial radial arterial segment, which is suitable for the measurement of radial wall elastance, is very small. Moreover, the amplitude of the sinusoidal displacement driven by the vibrator is very minute (0.15 mm). If the subject’s hand undergoing testing makes a trivial motion, a great response occurs due to the reactive force. Thus, in the study, we continuously registered several heartbeat cycles, and we chose the most ideal signal to assess the arterial wall elastance.

It is worth noting that perfectly locating the probe onto the radial artery is always a challenge. In this study, the measurement system allowed us to adjust the position of the moving probe. However, the probe tip could be properly located on the radial artery segment of interest through manual manipulation. During this adjustment, an appropriate location can be ultimately achieved by visually inspecting the reactive force waveform on the screen and through previous experience. It usually takes 1–2 min to complete the location procedure.

The frequency-dependent component of the arterial wall elastance increases with the vibration frequency according to Equation (7). According to the results, the wall elastance measured using the single-frequency approach with a lower frequency was close to the wall elastance measured using the multiple-frequency approach. As is known, the arterial wall elastance essentially varies with time. Thus, when the arterial wall is stimulated by a low-frequency vibration, an accurate wall elastance corresponding to a specific time cannot be yielded. This is why a very-low-frequency vibration cannot represent the time-varying wall elastance. A tradeoff problem, thus, appears between the selection of vibration frequency and the correct timing of wall elastance. According to our tradeoff tests, a vibration frequency of 40 Hz was used to measure the time-varying wall elastances. Figure 6, Figure 7 and Figure 8 show that the wall elastances measured using the 40 Hz approach were closest to the wall elastances measured using the multiple-frequency approach.

Arterial wall viscosity is related to energy dissipation and is regulated by the endothelium [34,35]. To eliminate the effect of viscosity on the wall elastance, we deliberately chose a specific time where peak deflection occurred in the sinusoidal displacement, D sin(ωt), and the value of cos(ωt) was equal to zero. Thus, the second term with cos(ωt) on the right side of Equation (5) could be ignored. That is, no viscosity effect existed when evaluating the arterial wall elastance in the present study.

A Bland–Altman plot (Figure 9) was employed to reveal the extent of agreement between the single-frequency and multiple-frequency vibration approaches in this study. The outliers within 1.0–1.4 of (E_single_ + E_multiple_)/2 originated from the cold-stress test. Under these conditions (4 °C), the radial arterial segment constricts through thermoregulation, resulting in a smaller diameter. Subsequently, it became more difficult to precisely locate the probe on the radial artery and to obtain accurate elastances measured using either the single- or the multiple-frequency approach. This may partially explain the existence of the outlier values in the Bland–Altman plot.

We found that the single-frequency approach proposed in this study has several advantages over the multiple-frequency approach. Firstly, since the mechanical vibration generated using a vibrator should have different frequencies when implementing the multiple-frequency method, its design is more complicated. Secondly, less computation is needed in the single-frequency method compared to the multiple-frequency approach in which a linear regression algorithm must be applied. Thirdly, the measurement time is shorter for the single-frequency method. Fourthly, since a 40 Hz displacement was adopted in the single-frequency approach, the effect of the frequency-dependent component on the wall elastance measurement was apparently small. In contrast, when a vibrator with different frequencies (from 40 to 85 Hz) was employed to press the arterial wall, a higher wall elastance in response to the higher-frequency stimulation emerged [36].

There were some limitations in the present work. Firstly, due to the low number of subjects in the present study, future studies with a larger cohort are still needed. Secondly, the proposed method is only available for assessing the wall elastance of a superficial arterial segment, such as the portion of the radial artery near the wrist, since the medium between the tip of the moving probe and the arterial wall may affect the measurement. Thirdly, the displacement or vibration used must be of a perfect sinusoidal waveform for the derivative of wall elastance to exclude the viscosity effect. Fourthly, the proposed method is sensitive to the measurement position. Lastly, the current study does not perform any comparative data analyses between the healthy young subjects and diseased patients. In the future, it will compare the arterial wall elastance measurement between healthy participants and age-matched patients with certain chronic disease, such as diabetes mellitus, hypertension, end-stage renal failure, and hyperlipidemia.

## 8. Conclusions

The proposed single-frequency approach combining a single-frequency vibrator and a force sensor was successfully applied to assess the time-varying wall elastance of radial artery. Furthermore, according to the point distribution in the Bland–Altman plot, there was good agreement between the proposed single-frequency approach and the existing multiple-frequency approach. Using the proposed method, a time-varying wall elastance curve corresponding to one cardiac cycle can be obtained. Therefore, this novel approach can be used for evaluating arterial stiffness to manage the risks of cardiovascular diseases in the future.

## Figures and Tables

**Figure 1 sensors-20-06463-f001:**
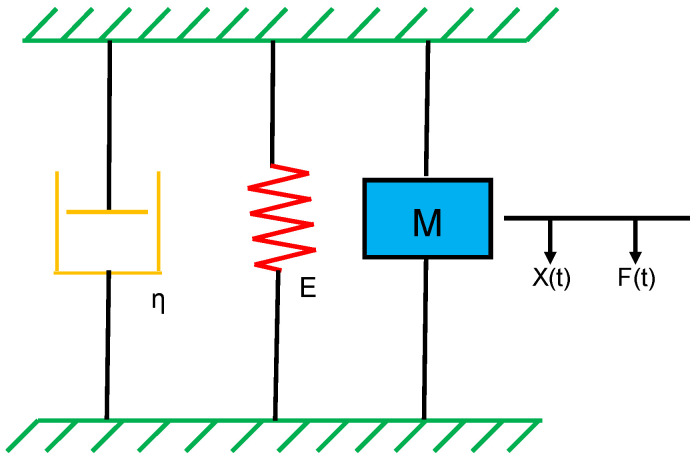
Schematic diagram of the vascular wall model with three components in parallel. η is the viscosity component, E is the elastance component, M is the effective mass, X(t) is the displacement of the vascular wall, and F(t) is the external force.

**Figure 2 sensors-20-06463-f002:**
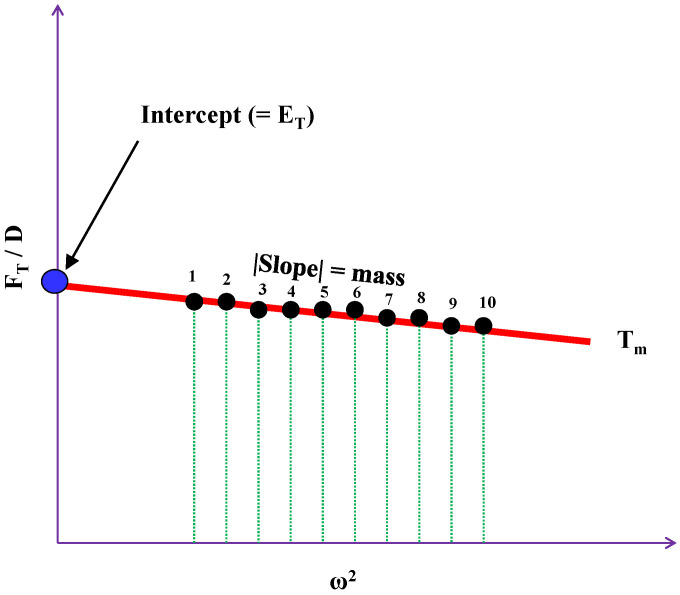
The relationship between F_T_/D and ω^2^ at a specific time, T_m_. The red line represents the regression line of the 10 points. E_T_ is the intercept of the vertical axis, and M is the absolute slope of the regression line.

**Figure 3 sensors-20-06463-f003:**
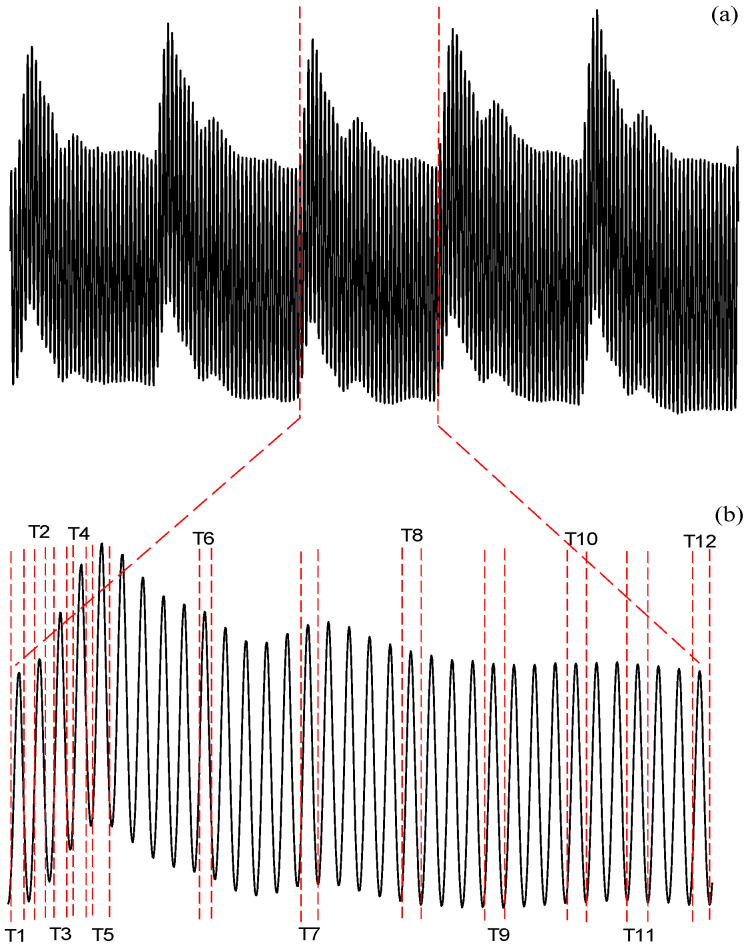
Explanation for yielding the peak reactive force at different timing points (T_1_–T_12_): (**a**) the reactive force signal of arterial wall’s response to a vibrator frequency of 40 Hz; (**b**) reactive force signal within one cardiac cycle with the 12 timings annotated.

**Figure 4 sensors-20-06463-f004:**
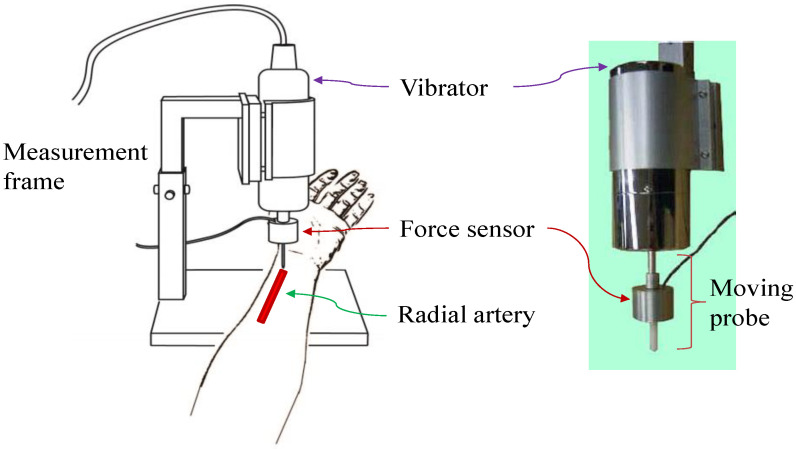
Schematic diagram of the measurement system. The vibrator drives the probe up and down with a specific frequency. The probe has a force sensor to detect the reactive force produced by the arterial wall.

**Figure 5 sensors-20-06463-f005:**
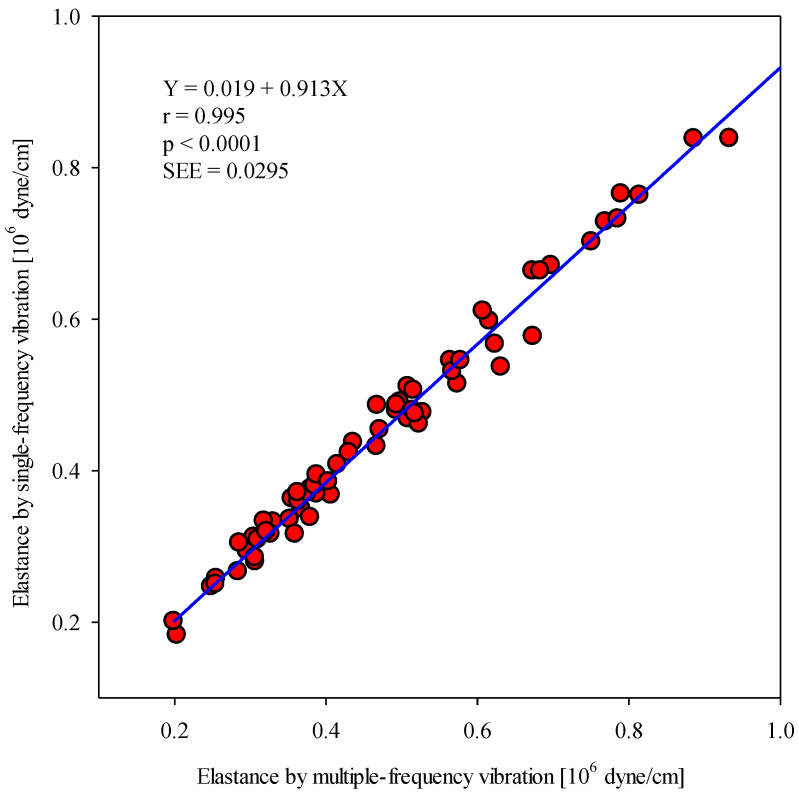
Linear relationship between maximum wall elastances measured using the multiple- and single-frequency vibration approaches under three thermal conditions (4 °C, 25 °C, and 42 °C).

**Figure 6 sensors-20-06463-f006:**
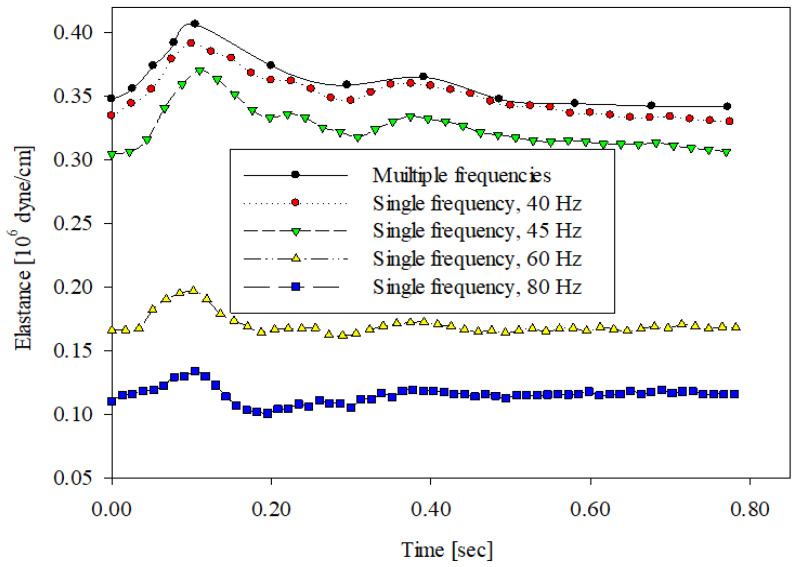
The time-varying wall elastances within a cardiac cycle measured using the multiple- and single-frequency approaches at room temperature (25 °C).

**Figure 7 sensors-20-06463-f007:**
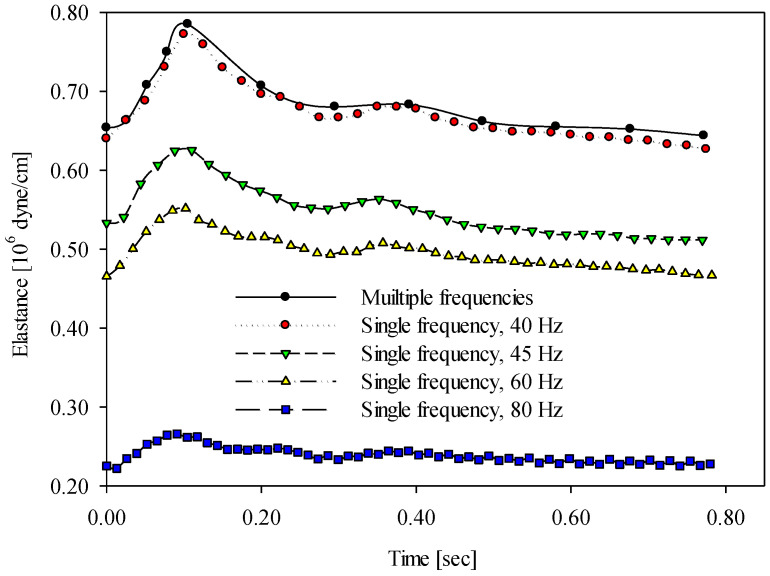
The time-varying wall elastances within a cardiac cycle measured using the multiple- and single-frequency approaches under cold stress (4 °C).

**Figure 8 sensors-20-06463-f008:**
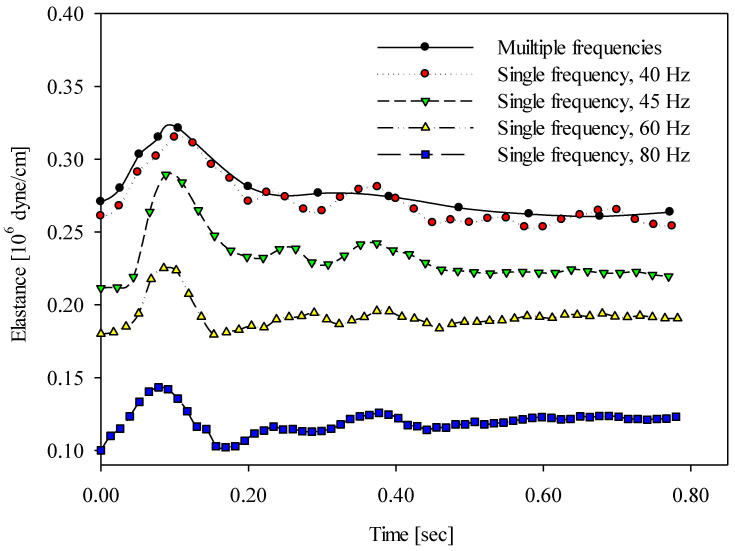
The time-varying wall elastances within a cardiac cycle measured using the multiple- and single-frequency approaches under hot stress (42 °C).

**Figure 9 sensors-20-06463-f009:**
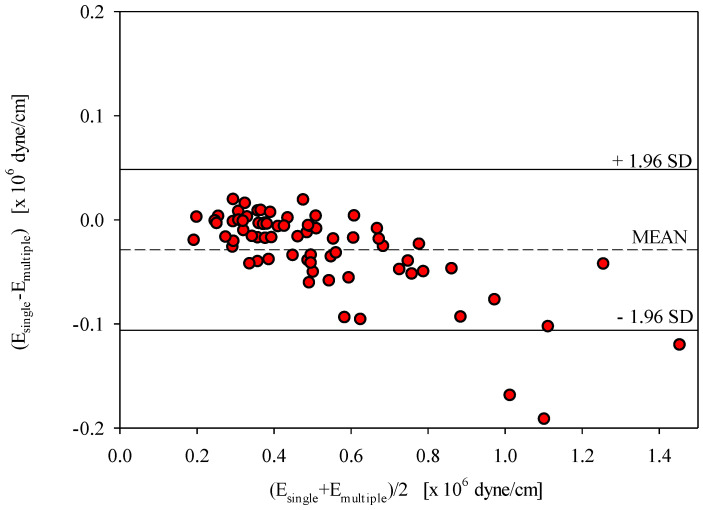
The precision and limits of agreement between E_single_ and E_multiple_.

**Table 1 sensors-20-06463-t001:** Comparison of E_multiple_ and E_single_ (10^3^ dyne/cm) at room temperature (25 °C), under cold stress (4 °C), and under hot stress (42 °C). Difference = absolute difference between E_multiple_ and E_single_ divided by E_multiple_ as a percentage.

Subject	Room Temperature (*N* = 28)			Cold Stress (*N* = 23)			Hot Stress (*N* = 21)		
	E_multiple_	E_single_	Difference	E_multiple_	E_single_	Difference	E_multiple_	E_single_	Difference
1	203	183	9.8	471	454	3.5	379	339	10.7
2	415	408	1.6	430	424	1.4	631	537	14.9
3	574	515	10.3	1012	935	7.6	305	312	2.6
4	673	664	1.3	683	664	2.7	379	376	0.8
5	307	280	8.6	623	567	9.0	508	469	7.7
6	367	350	4.7	751	703	6.4	310	309	0.2
7	564	546	3.3	578	546	5.6	321	320	0.4
8	330	333	0.8	385	381	1.1	364	360	1.0
9	406	368	9.4	785	733	6.7	-	-	-
10	296	294	0.6	362	371	2.5	-	-	-
11	255	258	1.3	616	598	2.9	-	-	-
12	248	247	0.5	284	267	5.9	-	-	-
13	327	316	3.2	1198	1007	16.0	364	360	1.0
14	492	480	2.5	514	480	6.7	-	-	-
15	790	766	3.0	607	611	0.6	-	-	-
16	355	363	2.4	523	462	11.6	199	201	1.3
17	318	334	4.9	388	395	1.8	352	336	4.5
18	886	839	5.3	-	-	-	527	477	9.6
19	1164	1061	8.8	-	-	-	814	764	6.1
20	306	285	6.8	403	386	4.3	359	316	11.8
21	436	437	0.4	517	475	8.1	933	839	10.0
22	567	531	6.3	-	-	-	467	432	7.4
23	769	729	5.2	1098	929	15.4	497	491	1.3
24	1514	1394	8.0	1778	1750	1.6	387	370	4.6
25	697	671	3.7	-	-	-	376	371	1.2
26	508	511	0.7	515	507	1.7	254	250	1.4
27	1277	1235	3.3	-	-	-	673	577	14.2
28	467	486	4.1	493	487	1.2	285	305	6.8
Mean ± STD	554 ± 324	532 ± 301	4.3 ± 3.1	653 ± 339 **	641 ± 325 **	5.4 ± 4.4	444 ± 185 *	417 ± 158 *	5.4 ± 4.8

* *p* < 0.05 versus room temperature; ** *p* < 0.001 versus room temperature.

**Table 2 sensors-20-06463-t002:** Linear relationship between elastances measured using the multiple- and single-frequency vibration methods at room temperature (25 °C), under cold stress (4 °C), and under hot stress (42 °C). Multiple frequencies range from 40 to 85 Hz with a step of 5 Hz; single frequency = 40 Hz. SEE, standard error of the estimate.

Condition	Linear Regression	Correlation Coefficient, *r*	SEE	*p*-Value
Room temperature (25 °C) (*N* = 28)	E_s__ingle_ = 0.021 + 0.922E_m__ultiple_	0.998	0.0198	<0.0001
Cold stress (4 °C) (*N* = 23)	E_s__ingle_ = 0.017 + 0.915E_m__ultiple_	0.991	0.0423	<0.0001
Hot stress (42 °C) (*N* = 21)	E_s__ingle_ = 0.041 + 0.846E_m__ultiple_	0.993	0.0198	<0.0001

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
