# Peer review of "Noninvasive Measurement of Time-Varying Arterial Wall Elastance Using a Single-Frequency Vibration Approach"

_sensors, 2020, doi:10.3390/s20226463_

Round 1

Reviewer 1 Report

The study was of interest and the methodology to assess arterial conditions is acceptable. On the other hand, this study is still preliminary, and the paper’s quality is expected to be more raised. The authors must sincerely address the below points:

  1. We want to see the comparative data between healthy and diseased populations, because we have had the previous experience such that the measurement was feasible but the clinical relevance was not fully clear. This is important for the significance of the apparatus. Furthermore, a translational study is required to be published internationally and also in this Journal.
  2. We want to see the comparative data between the present apparatus and other popular apparatus (e.g., PWV) in the same population. It is important to show the superiority of the present apparatus to the already used apparatus.
  3. We need more in-detailed description on the way how to look for (locate) precisely the suitable radial arteries for measurement. This is important for the application of the apparatus to any subjects measured.
  4. The references cited were required in the tests of cold and hot stress. This was important for the comparison to the results of other apparatus.
  5. We want to know the reason why the number of the studied subjects was different as table 1 and 2).
  6. The histogram of data by the single-frequency approach in the respective condition can be added for Table 1 (not only mean plus SD but also the histogram is important information in this study).
  7. We want to know that the present apparatus is easily made and the multiple production is possible. If so, the correlation and difference in the data between the apparatuses can be added.
  8. The comparatively outlier values of some cases (around the high [1.0-1.4] of [(E single+E multiple)/2]) were seen. We need more description on the interpretation. This may be related to the future data when the authors study the diseased subjects.
  9. The English can be more improved in the use of indefinite and definite articles.

Thanks for the opportunity to review the nice innovative paper.

Author Response

To Reviewer #1:

Thank the first reviewer for his/her valuable comments that make better this manuscript. The texts in this revised manuscript have been corrected/ modified by red words.

The study was of interest and the methodology to assess arterial conditions is acceptable. On the other hand, this study is still preliminary, and the paper’s quality is expected to be more raised. The authors must sincerely address the below points:

  1. We want to see the comparative data between healthy and diseased populations, because we have had the previous experience such that the measurement was feasible but the clinical relevance was not fully clear. This is important for the significance of the apparatus. Furthermore, a translational study is required to be published internationally and also in this Journal.

Ans: The present study mainly focuses on the feasibility of the proposed single-frequency approach. Therefore, the authors faithfully demonstrate comparative data of arterial elastance measured with both the single- and multiple-frequency approaches under three thermal conditions in the study.

In Introduction

Most of the techniques mentioned above can roughly provide an index of arterial stiffness for either an arterial segment or a whole arterial system. However, the arterial characteristics, especially in terms of arterial compliance and wall elastance, are not constant in essence but time-varying and dependent on the transmural pressure [8,17,22]. The Voigt and Maxwell model has been used to measure the time-varying arterial wall elastance with a multiple-frequency vibrator approach [17]. Because our method measures the time-varying arterial wall elastance with a simplified Voigt and Maxwell model [25], the measured elastances were compared with those obtained using the multiple-frequency vibration approach. Different vibrational frequencies were employed to evaluate the performance of this proposed approach. Moreover, to intentionally alter the arterial wall elastance, we measured the arterial wall elastance under cold- and hot-stress trials. Twenty-eight healthy and young subjects were recruited to participate in these experiments. It was found that a lower vibrational frequency resulted in a closer arterial wall elastance measured using the multiple-frequency vibrator approach. Therefore, the purposes of our study were (1) to present the derivation of a new method for measuring arterial wall elastance with a single-frequency vibration, (2) to compare the elastance values measured with the single- and the multiple-frequency vibration approaches in three thermal conditions, and (3) to validate the feasibility of the proposed approach by comparing its precision and agreement with the multiple-frequency approach.

  1. We want to see the comparative data between the present apparatus and other popular apparatus (e.g., PWV) in the same population. It is important to show the superiority of the present apparatus to the already used apparatus.

Ans: In the present study, we do not measure the PWV of subjects. Basically, both Esingle and Emultiple belong to local elastances, and PWV (e.g. from aorta to radial artery) is a stiffness index of a long arterial segment. In the present study, the effects of the cold stress and hot stress are only on a part/segment of the superficial radial artery near the wrist. These thermal stress stimulations may not directly alter the PWV, as compared with the room temperature condition. However, the valuable comment indicates a promising direction in future application studies using the proposed approach.     

  1. We need more in-detailed description on the way how to look for (locate) precisely the suitable radial arteries for measurement. This is important for the application of the apparatus to any subjects measured.

Ans: It is always one of challenges to perfectly position the probe onto the superficial site of the radial artery near the wrist.  In the measuring system, the moving probe allows us to move up and down. However, the probe tip is properly located on the radial artery segment of interest through manual manipulation. During the manual adjustment procedure, we can achieve an appropriate location both by visually inspecting the reactive force waveform on the screen and by our previous experience. It usually takes 1~2 minutes to perform the location procedure.  The authors have added several sentences to discuss this locating challenge in Discussion section.

In Discussion

It is worth noting that perfectly locating the probe onto the radial artery is always a challenge. In this study, the measurement system allowed us to adjust the position of the moving probe. However, the probe tip could be properly located on the radial artery segment of interest through manual manipulation. During this adjustment, an appropriate location can be ultimately achieved by visually inspecting the reactive force waveform on the screen and through previous experience. It usually takes 1–2 min to complete the location procedure.

  1. The references cited were required in the tests of cold and hot stress. This was important for the comparison to the results of other apparatus.

Ans:  Some investigations related to the cold and hot stress have been cited in the reference list.  The authors have added one paragraph to discuss and compare the present results with previous studies under cold or hot stimulations.    

In Discussion

Skin circulation is controlled by thermoregulatory and nonthermoregulatory reactions [30,31]. Thermoregulatory reactions include skin blood flow and local arterial compliance responses to heat and cold stresses. Two branches of the sympathetic nervous system exert these effects, a noradrenergic vasoconstrictor branch and an active vasodilator branch. A recent study showed that cold stress causes acute decreases in central and peripheral compliance [32], whereas heat stress increases vascular compliance [33]. Therefore, in this study, cold stress and hot stress were chosen to induce an alteration in the radial arterial wall elastances. The regression line between Esingle and Emultiple obtained under three different thermal conditions (4 °C, 25 °C, and 42 °C) was found to have a high correlation coefficient of 0.995, as shown in Figure 5. In addition, the distribution of 72 scattered points in the Bland–Altman plot in Figure 9 shows that most points were within the limits of agreement, suggesting good agreement between the single-frequency and multiple-frequency approaches. Moreover, in Table 1, Esingle under both cold stress and hot stress showed significant differences with respect to that at room temperature. Moreover, among the three thermal conditions, the average of Esingle under cold stress was largest, and the average of Esingle under hot stress was smallest. These findings are consistent with physiological thermal stimulations.

  1. Roca, F.; Bellien, J.; Iacob, M.; Joannides, R. Endothelium-dependent adaptation of arterial wall viscosity during blood flow increase is impaired in essential hypertension. Atherosclerosis. 2019, 285, 102-107, doi: 10.1016/j.atherosclerosis.2019.04.208.
  2. Roca, F.; Iacob, M.; Remy-Jouet, I.; Bellien, J.; Joannides, R. Evidence for a role of vascular endothelium in the control of arterial wall viscosity in humans. Hypertension. 2018, 71, 143-150.
  3. Chen, X.; Sala-Mercado, J.A.; Hammond, R.L.; Ichinose, M.; Soltani, S.; Mukkamala, R.; O'Leary, D.S.; Dynamic control of maximal ventricular elastance via the baroreflex and force-frequency relation in awake dogs before and after pacing-induced heart failure. J. Physiol. Heart. Circ. Physiol. 2010, 299, H62-69.
  4. Ganio, M.S.; Brothers, R.M.; Shibata, S.; Hastings, J.L.; Crandall, C.G. Effect of passive heat stress on arterial stiffness. Physiol. 2011, 96, 919-926.

  1. We want to know the reason why the number of the studied subjects was different as table 1 and 2).

Ans: The subjects recruited all are from the college. Some of them must leave for their classes. So, the subject number in the cold or hot stress is less than that in the room temperature condition.    

  1. The histogram of data by the single-frequency approach in the respective condition can be added for Table 1 (not only mean plus SD but also the histogram is important information in this study).

Ans: The authors have modified the content of Table 1 according this valuable comment.

In Results

Table 1. Comparison of Emultiple and Esingle (103 dyne/cm) at room temperature (25 °C), under cold stress (4 °C), and under hot stress (42°C). Difference = absolute difference between Emultiple and Esingle divided by Emultiple as a percentage.

Subject

Room temperature (N=28)

Cold stress (N=23)

Hot stress (N=21)

Emultiple

Esingle

Difference  

Emultiple

Esingle

Difference

Emultiple

Esingle

Difference

1

203

183

9.8

471

454

3.5

379

339

10.7

2

415

408

1.6

430

424

1.4

631

537

14.9

3

574

515

10.3

1012

935

7.6

305

312

2.6

4

673

664

1.3

683

664

2.7

379

376

0.8

5

307

280

8.6

623

567

9.0

508

469

7.7

6

367

350

4.7

751

703

6.4

310

309

0.2

7

564

546

3.3

578

546

5.6

321

320

0.4

8

330

333

0.8

385

381

1.1

364

360

1.0

9

406

368

9.4

785

733

6.7

x

x

x

10

296

294

0.6

362

371

2.5

x

x

x

11

255

258

1.3

616

598

2.9

x

x

x

12

248

247

0.5

284

267

5.9

x

x

x

13

327

316

3.2

1198

1007

16.0

364

360

1.0

14

492

480

2.5

514

480

6.7

x

x

x

15

790

766

3.0

607

611

0.6

x

x

x

16

355

363

2.4

523

462

11.6

199

201

1.3

17

318

334

4.9

388

395

1.8

352

336

4.5

18

886

839

5.3

x

x

x

527

477

9.6

19

1164

1061

8.8

x

x

x

814

764

6.1

20

306

285

6.8

403

386

4.3

359

316

11.8

21

436

437

0.4

517

475

8.1

933

839

10.0

22

567

531

6.3

x

x

x

467

432

7.4

23

769

729

5.2

1098

929

15.4

497

491

1.3

24

1514

1394

8.0

1778

1750

1.6

387

370

4.6

25

697

671

3.7

x

x

x

376

371

1.2

26

508

511

0.7

515

507

1.7

254

250

1.4

27

1277

1235

3.3

x

x

x

673

577

14.2

28

467

486

4.1

493

487

1.2

285

305

6.8

Mean

±STD

554

±324

532 ±301

4.3±3.1

653 ±339**

641 ±325**

5.4±4.4

444 ±185*

417 ±158*

5.4±4.8

*: p<0.05 versus Room Temperature, **: p<0.001 versus Room Temperature.

  1. We want to know that the present apparatus is easily made and the multiple production is possible. If so, the correlation and difference in the data between the apparatuses can be added.

Ans: In the study, we use the same apparatus to generate the minute mechanical vibration. Although the present apparatus, the vibrator (DPS-270, DiaMidical System Cor., Tokyo, Japan), can provide minute vibration with different frequencies, it is big and heavy and complex. We show the difference in the elastance data between single- and multiple-frequency approaches in Table 1. Their correlations are shown in Table 2 and Figure 5.

Table 2 shows the correlation between Esingle and Emultiple. For all three temperatures (4 °C, 25 °C, and 42 °C), very high correlation coefficients (larger than 0.99) were found between Esingle and Emultiple. Furthermore, we found that the slopes of the three regression lines corresponding to the different temperatures were all smaller than 1.0. Combining all values of Esingle and Emultiple measured at the three different temperatures, we established a regression line (Esingle = 0.019 + 0.913Emultiple) with a Pearson correlation coefficient of 0.995, as shown in Figure 5.

Table 2. Linear relationship between elastances measured using the multiple- and single-frequency vibration methods at room temperature (25 °C), under cold stress (4 °C), and under hot stress (42 °C). Multiple frequencies range from 40 to 85 Hz with a step of 5 Hz; single frequency = 40 Hz. SEE, standard error of the estimate.

Condition

Linear regression

Correlation coefficient, r

SEE

p-Value

Room temperature (25 °C) (N = 28)

Esingle = 0.021 + 0.922Emultiple

0.998

0.0198

<0.0001

Cold stress (4 °C)

(N = 23)

Esingle = 0.017 + 0.915Emultiple

0.991

0.0423

<0.0001

Hot stress (42 °C)

(N = 21)

Esingle = 0.041 + 0.846Emultiple

0.993

0.0198

<0.0001

Figure 5. Linear relationship between maximum wall elastances measured using the multiple- and single-frequency vibration approaches under three thermal conditions (4 °C, 25 °C, and 42 °C).

8.The comparatively outlier values of some cases (around the high [1.0-1.4] of [(E single+E multiple)/2]) were seen. We need more description on the interpretation. This may be related to the future data when the authors study the diseased subjects.

Ans: Most pared points within [1.0-1.4] of [(Esingle + Emultiple)/2]) are from the cold stress. In the cold stimulation (4°C), the radial arterial segment may constrict via the so-called thermal regulation, resulting in a smaller diameter.

Subsequently, it becomes difficult to precisely locate the probe on the radial artery and to obtain accurate elastances measured with either the single- or the multiple-frequency approach. One paragraph has been added to discuss the outlier phenomenon in Discussion section.

In Discussion

A Bland–Altman plot (Figure 9) was employed to reveal the extent of agreement between the single-frequency and multiple-frequency vibration approaches in this study. The outliers within 1.0–1.4 of (Esingle + Emultiple)/2 originated from the cold-stress test. Under these conditions (4 °C), the radial arterial segment constricts through thermoregulation, resulting in a smaller diameter. Subsequently, it became more difficult to precisely locate the probe on the radial artery and to obtain accurate elastances measured using either the single- or the multiple-frequency approach. This may partially explain the existence of the outlier values in the Bland–Altman plot.  

9.The English can be more improved in the use of indefinite and definite articles.

Ans: The revised manuscript has been sent to MDPI Author Services for English editing. So, the writing quality of the text in the revised manuscript is significantly improved.  

Reviewer 2 Report

All the work is well done, well written and the subject matter is of great relevance.
I would like the authors to discuss the negative bias for values higher than 0.8 10^6 dyne/cm highlighted in the systematic deviation (figure 9) The precision and the limits of agreement between Esingleand Emultiple.
Justify the choice of your parametric statistics.

Author Response

To Reviewer #2:

Thank the second reviewer for his/her valuable comments that make better this manuscript. The texts in this revised manuscript have been corrected/ modified by red words.

  1. All the work is well done, well written and the subject matter is of great relevance.
    I would like the authors to discuss the negative bias for values higher than 0.8 10^6 dyne/cm highlighted in the systematic deviation (Figure 9) The precision and the limits of agreement between Esingle and Emultiple.

Ans: Most pared points within [1.0-1.4] of [(Esingle + Emultiple)/2]) are from the cold stress. In the cold stimulation (4°C), the radial arterial segment may constrict via the so-called thermal regulation, resulting in a smaller diameter. Subsequently, it becomes difficult to precisely locate the probe on the radial artery and to obtain accurate elastances measured with either the single- or the multiple-frequency approach. One paragraph has been added to discuss the outlier phenomenon in Discussion section.

In Discussion

A Bland–Altman plot (Figure 9) was employed to reveal the extent of agreement between the single-frequency and multiple-frequency vibration approaches in this study. The outliers within 1.0–1.4 of (Esingle + Emultiple)/2 originated from the cold-stress test. Under these conditions (4 °C), the radial arterial segment constricts through thermoregulation, resulting in a smaller diameter. Subsequently, it became more difficult to precisely locate the probe on the radial artery and to obtain accurate elastances measured using either the single- or the multiple-frequency approach. This may partially explain the existence of the outlier values in the Bland–Altman plot.  

  1. Justify the choice of your parametric statistics.

Ans: All the sample sizes of the room-temperature, cold and hot groups are less than 30 in the study. That is why we choose a two-tailed paired t-test to compare their difference in mean between two groups. Also, the difference percentage between all paired values measured with the single- and the multiple-frequency approach is calculated to evaluate the feasibility of the proposed method. Here, the difference percentage (%) is defined as the absolute difference between Emultiple and Esingle divided by Emultiple .

In Results

Table 1. Comparison of Emultiple and Esingle (103 dyne/cm) at room temperature (25 °C), under cold stress (4 °C), and under hot stress (42°C). Difference = absolute difference between Emultiple and Esingle divided by Emultiple as a percentage.

Subject

Room temperature (N=28)

Cold stress (N=23)

Hot stress (N=21)

Emultiple

Esingle

Difference  

Emultiple

Esingle

Difference

Emultiple

Esingle

Difference

1

203

183

9.8

471

454

3.5

379

339

10.7

2

415

408

1.6

430

424

1.4

631

537

14.9

3

574

515

10.3

1012

935

7.6

305

312

2.6

4

673

664

1.3

683

664

2.7

379

376

0.8

5

307

280

8.6

623

567

9.0

508

469

7.7

6

367

350

4.7

751

703

6.4

310

309

0.2

7

564

546

3.3

578

546

5.6

321

320

0.4

8

330

333

0.8

385

381

1.1

364

360

1.0

9

406

368

9.4

785

733

6.7

x

x

x

10

296

294

0.6

362

371

2.5

x

x

x

11

255

258

1.3

616

598

2.9

x

x

x

12

248

247

0.5

284

267

5.9

x

x

x

13

327

316

3.2

1198

1007

16.0

364

360

1.0

14

492

480

2.5

514

480

6.7

x

x

x

15

790

766

3.0

607

611

0.6

x

x

x

16

355

363

2.4

523

462

11.6

199

201

1.3

17

318

334

4.9

388

395

1.8

352

336

4.5

18

886

839

5.3

x

x

x

527

477

9.6

19

1164

1061

8.8

x

x

x

814

764

6.1

20

306

285

6.8

403

386

4.3

359

316

11.8

21

436

437

0.4

517

475

8.1

933

839

10.0

22

567

531

6.3

x

x

x

467

432

7.4

23

769

729

5.2

1098

929

15.4

497

491

1.3

24

1514

1394

8.0

1778

1750

1.6

387

370

4.6

25

697

671

3.7

x

x

x

376

371

1.2

26

508

511

0.7

515

507

1.7

254

250

1.4

27

1277

1235

3.3

x

x

x

673

577

14.2

28

467

486

4.1

493

487

1.2

285

305

6.8

Mean

±STD

554

±324

532 ±301

4.3±3.1

653 ±339**

641 ±325**

5.4±4.4

444 ±185*

417 ±158*

5.4±4.8

*: p<0.05 versus Room Temperature, **: p<0.001 versus Room Temperature.

  1. Statistical Analysis

The quantitative data are expressed as the mean ± SD. A two-tailed paired t-test was used to compare the average of the maximum wall elastances in cold stress or hot stress with that at room temperature. A p-value of 0.05 or less was considered statistically significant. The maximum values of time-varying wall elastances measured using the multiple- and the single-frequency approaches are represented by Emultiple and Esingle, respectively. The degree of linear relationship between Esingle and Emultiple was expressed with a Pearson correlation coefficient using Sigma Plot 12.0 (Systat Software, Inc., USA). The percentage difference between all paired values measured using these two approaches was calculated to further evaluate the feasibility of the proposed method. Here, the percentage difference (%) was defined as the absolute difference between Emultiple and Esingle divided by Emultiple. Furthermore, the precision of and the agreement between Esingle and Emultiple were compared using a Bland–Altman plot [27].

Round 2

Reviewer 1 Report

The paper has been much improved. However, many readers will hope seeing the comparative data between the healthy young students and diseased subjects. In the paragraph of the study limitations (discussion part), it should be clearly and concretely stated. What/which diseases will be examined in the near future?

Author Response

To Reviewer #1:

Thank the first reviewer for his/her valuable comments that make better this manuscript. The texts in this revised manuscript have been corrected/ modified by red words.

The paper has been much improved. However, many readers will hope seeing the comparative data between the healthy young students and diseased subjects. In the paragraph of the study limitations (discussion part), it should be clearly and concretely stated. What/which diseases will be examined in the near future?

Ans: Thank you for your valuable comment. The authors have modified the paragraph of the study limitations in Discussion section.

In Discussion

There were some limitations in the present work. Firstly, due to the low number of subjects in the present study, future studies on a larger cohort are still needed. Secondly, the proposed method is only available for assessing the wall elastance of a superficial arterial segment, such as the portion of the radial artery near the wrist, since the medium between the tip of the moving probe and the arterial wall may affect the measurement. Thirdly, the displacement or vibration used must be of a perfect sinusoidal waveform for the derivative of wall elastance to exclude the viscosity effect. Fourthly, the proposed method is sensitive to the measurement position. Lastly, the current study does not perform any comparative data between the healthy young subjects and diseased patients. In the future, it will compare the arterial wall elastance measurement between healthy participants and age-matched patients with certain chronic disease, such as diabetes mellitus, hypertension, end-stage renal failure, and hyperlipidemia.
